# An Exceedingly Rare Case of Mechanobullous Epidermolysis Bullosa Acquisita in a Prepubertal Child: A Review of the Clinical and Laboratory Considerations

**DOI:** 10.3390/antib14020034

**Published:** 2025-04-11

**Authors:** Aleksandra Wiktoria Bratborska, Maciej Spałek, Monika Bowszyc-Dmochowska, Marian Dmochowski

**Affiliations:** 1Department of Dermatology, Poznan University of Medical Sciences, 60-355 Poznan, Poland; aleksandrabratborska@gmail.com; 2Doctoral School, Poznan University of Medical Sciences, 61-701 Poznan, Poland; 3Autoimmune Blistering Dermatoses Section, Department of Dermatology, Poznan University of Medical Sciences, 60-355 Poznan, Poland; mspalek@ump.edu.pl; 4Cutaneous Histopathology and Immunopathology Section, Department of Dermatology, Poznan University of Medical Sciences, 60-355 Poznan, Poland; mbowdmo@ump.edu.pl

**Keywords:** epidermolysis bullosa acquisita, immunological diagnostics, autoimmunity

## Abstract

Introduction: Epidermolysis bullosa acquisita (EBA) is a rare autoimmune disease causing subepithelial blistering due to autoantibodies against type VII collagen. While mechanobullous EBA predominantly affects adults, our report presents an exceedingly rare case in an 11-year-old football player. Case Report: The patient reported a one-year history of blistering and scarring on the knees and scrotum. The diagnosis was established with direct immunofluorescence (DIF), mosaic indirect immunofluorescence (IIF) showing IgG antibodies reacting with the dermal side of salt-split primate skin, and multiplex ELISA revealing an elevated level of IgG antibodies against type VII collagen. Treatment with a superpotent topical glucocorticosteroid and activity modifications improved his condition. Review: This case highlights the importance of considering EBA in differential diagnoses of pediatric blistering diseases and suggests that conservative management may be effective in mild cases. We also review clinical and laboratory considerations on the topic of childhood EBA. Conclusions: Further studies are essential to develop evidence-based guidelines for pediatric EBA.

## 1. Introduction

Epidermolysis bullosa acquisita (EBA) is a chronic autoimmune mucocutaneous bullous disease (AIBD) characterized by subepithelial blisters and subsequent scarring [1]. EBA is associated with IgG or, less commonly, IgA autoantibodies, usually against the non-collagenous domains (NC1 and NC2) of type VII collagen, an essential component of the anchoring fibrils securing the epithelial basement membrane to the underlying tissue [2]. EBA appears to be the rarest subepidermal blistering disease, exhibiting a prevalence of fewer than 0.5 cases per million individuals [3]. It predominantly affects adults, and only 40 cases in children have been documented [4].

EBA presents in two primary clinical variants: the inflammatory vesiculobullous phenotype and the mechanobullous form. The inflammatory phenotype, while more common, presents in four clinical subtypes: Bullous Pemphigoid-Like, Mucous Membrane Pemphigoid-Like, Linear IgA Disease-Like, and Brunsting–Perry Pemphigoid-Like [1,5]. The Bullous Pemphigoid-Like variant is characterized by tense bullae, erosive lesions, crusting, and urticaria-like erythema, frequently accompanied by pruritus. The lesion distribution typically involves the trunk, skin folds, extensor surfaces of the limbs, and distal extremities, with sporadic facial involvement. Mucosal membrane infiltration is uncommon, and erosions tend to heal with atrophic scarring and milium formation. This variant constitutes approximately 25–50% of all cases [6]. The Mucous Membrane Pemphigoid-Like variant entails significant mucosal involvement and is associated with complications such as dysphagia and malnutrition, necessitating a multidisciplinary and aggressive therapeutic approach [7,8]. The Linear IgA Disease-Like variant is distinguished by tense vesicles, erythematous lesions, and annular or polycyclic plaques, characterized by vesicles at the periphery, known as the “string of pearls” sign. Notably, this variant lacks prominent scarring or milia and exhibits infrequent mucosal involvement, occurring in about 4% of cases [9]. The Brunsting–Perry Pemphigoid-Like variant displays subepidermal blistering, primarily affecting the head and neck region, and is characterized by atrophic scars without any mucosal involvement [1].

The mechanobullous variant involves non-inflammatory, trauma-induced blistering predominantly on extensor surfaces, often progressing to scarring, milium formation, and nail dystrophy. This variant is exceedingly rare in children, with only seven cases reported to date [4].

## 2. Case Report

We report a case of an 11-year-old prepubertal male, an active football player, presenting with barely itching cutaneous lesions, namely chronic blistering, erosions, and scarring. The patient had a one-year history of skin lesions on the knees and scrotum. A physical examination revealed erythematous, exfoliative lesions on the knees (Figure 1) and a solitary blister on the scrotum, accompanied by evident scarring and crusting (Figure 2). The oral mucosa was free of lesions, and the patient did not report any gastrointestinal symptoms.

Direct immunofluorescence (DIF) of the perilesional scrotal skin demonstrated linear C3 (+++) (Figure 3) but not IgG, IgG1, IgG4, IgA, or IgM deposits along the dermal–epidermal junction in an undetermined pattern. Mosaic indirect immunofluorescence (IIF) (Euroimmun, Lübeck, Germany) on salt-split primate skin revealed IgG antibodies localized to the dermal side of the split, with a titer of 1:10. Furthermore, multiplex ELISA (Euroimmun, Germany) detected an elevated level of IgG antibodies against type VII collagen (2.55, with a cut-off value of 1). A serological ELISA test (Euroimmun, Lübeck, Germany) was negative for IgA antibodies against tissue transglutaminase (the level was below 0.6 RU/mL; the cut-off level is 20.0 RU/mL).

Based on these findings, a diagnosis of mechanobullous EBA was established, and superpotent topical glucocorticosteroid therapy with clobetasol propionate in the ointment was initiated on the affected skin lesions. We recommended administering the treatment twice a day until the complete healing of the lesions. Also, the patient was advised to avoid high-impact activities, such as football, to minimize trauma-induced blistering. Unfortunately, the patient was then lost to follow-up.

## 3. Detailed Descriptions of Crucial Laboratory Procedures to Diagnose EBA

### 3.1. Direct Immunofluorescence Procedure

DIF [10] of perilesional tissue was performed in this case for the detection of not only the conventionally evaluated IgA, IgM, IgG, and C3 deposits but also IgG1 and IgG4 deposits. The experience of our laboratory and other laboratories shows that images of IgG4 deposition can be more suitable for unequivocal interpretation than images of standard immunoreactants, including IgG [11,12]. The absence of detectable immunoglobulins in the DIF analysis of our patient’s skin sample may be attributable to their deposition levels falling below the sensitivity threshold. For DIF staining, 4 μm cryostat sections of perilesional skin were cut. The tissue sections were incubated in a humid chamber for 30 min at room temperature (RT) with commercially available fluorescein isothiocyanate (FITC)-conjugated rabbit polyclonal antibodies against human IgA, IgM, IgG (code No. F 0202), and C3 (Dako, Denmark) and FITC-conjugated mouse monoclonal antibodies against human IgG subclasses IgG1 and IgG4 (clone HP-6025) (Sigma-Aldrich, St. Louis, MO, USA). The antibodies were used at a working dilution of 1: 100 in phosphate buffer saline (PBS). The samples were then washed in PBS (pH 7.2) at RT for 15 min with gentle agitation. Then, slides were coverslipped and examined by two independent observers to minimize the subjectivity of the evaluation. A short-arc mercury lamp-operated microscope (BX40, Olympus, Japan) was used to visualize DIF images. The intensities of deposits on the slides were graded using a semiquantitative 4-point scale (from “−” to “+++”) at the original objective magnification of 40×.

### 3.2. Multiplex ELISA Procedure

The commercially available multiplex ELISA system [10] developed in Lübeck, Germany, which incorporates a multi-center approach for validation, was applied to detect serum IgG antibodies in a single procedure [13,14]. This system comprises different dominantly immunogenic epitopes of 6 antigens (BP180, BP230, DSG1, DSG3, envoplakin, NC1 domain of type VII collagen). In the multiplex ELISA, each antigen was coated in a separate well, and a semiquantitative evaluation was carried out using the manufacturer’s cut-off ratio of 1. All measurements were made in the ELISA plate readers (Asys Expert 96 or Ledetect 96) equipped with MikroWin 2000 software by a single operator following the manufacturer’s instructions.

## 4. Review of Clinical and Laboratory Considerations

Despite its rarity, childhood EBA should be considered in the differential diagnosis of blistering diseases in pediatric patients. We now advocate the three-tier approach for diagnosing AIBDs: evaluating clinical features, imaging the DIF of perilesional tissue, and performing biochemical–molecular serum studies to identify immunogenic protein epitopes targeted by an autoimmune response. Thus, the EBA minimal and sufficient laboratory diagnostic examinations should be DIF and multiplex ELISA containing type VII collagen. With this approach, the mechanobullous and inflammatory phenotypes of IgG-mediated EBA should no longer be a diagnosis of exclusion [15]. The detection of only C3 deposits in DIF, as in our case, should be interpreted cautiously because C3 is activated via the classical, alternative, or lectin pathway [16]. Activation of the lectin pathway, unlike that of the classic and alternative pathways, is not initiated by an antibody response. Therefore, isolated C3 deposits do not necessarily indicate the autoimmune nature of the disease. In our case, the detection of serum IgG antibodies to type VII collagen demonstrated that the autoimmune response indeed mediated the disease. Our patient had no mucosal lesions, although most cases of childhood EBA report mucosal involvement.

A biopsy of the lesional tissue for hematoxylin and eosin (HE) histopathology was not performed, as it fails to reveal the immunopathological nature of the disease, showing only non-specific findings such as subepidermal blistering and neutrophil-dominant dermal inflammatory infiltrate. To minimize additional trauma to the sensitive scrotal skin in pediatric patients, we opt for a DIF technique specifically tailored for the detection of autoimmune blistering diseases.

Dermatitis herpetiformis, which must be considered after the evaluation of clinical features, was excluded due to the normal levels of IgA antibodies against tissue transglutaminase in the immunopathological assessment. Notably, pediatric patients suffering from IgA-EBA and presenting with erythematous arciform cutaneous lesions accompanied by a limited number of scars and milium cysts may be misdiagnosed as having linear IgA disease [17]. As the skin lesions in our patient were limited to the skin on the knee and scrotum, a topical therapy combined with lifestyle modification to avoid trauma seemed appropriate to achieve a satisfying improvement in the dermatological state. However, in more severe cases, in which blisters affect larger skin areas, as well as mucous membranes, systemic therapy is recommended. The available literature reports successful outcomes after treatment with a combination of prednisone at 1 mg/kg body weight and dapsone at 50 mg daily, or after monotherapy with dapsone at 100 mg daily [18,19]. While adult EBA is associated with frequent resistance to treatment, pediatric EBA has a more favorable prognosis [20]. In adult populations, the reactivity of autoantibodies is primarily limited to the NC1 domain of collagen type VII [21]. In contrast, Japanese pediatric patients diagnosed with EBA were found to have autoantibodies against epitopes located in the triple-helical region and not the NC1 domain [22]. In addition, a 4-year-old female EBA patient exhibited IgG autoantibodies that reacted with the NC1, NC2, and triple-helical domains of type VII collagen [23]. Importantly, the tissue-bound and circulating autoantibodies specifically targeting the triple-helical central domain are linked to the less severe clinical presentation of the disease in children [24,25]. It is therefore possible that the autoimmune response in individual patients with childhood EBA changes over the course of the treatment-naïve disease. From our experience, the inflammatory phenotype has a better long-term prognosis.

Notably, the co-occurrence of EBA with other autoimmune diseases, such as inflammatory bowel disease (IBD), rheumatoid arthritis, and chronic thyroiditis, among others, suggests a need for clinicians to be vigilant in monitoring for these conditions [1]. The potential coexistence of EBA and IBD can be attributed to the expression of type VII collagen in the junction between the intestinal epithelium and the lamina propria [26]. Autoimmune diseases share genetic and immunologic pathways, and the association of EBA with specific human leukocyte antigen (HLA) haplotypes, such as HLA-DRB1*13 and HLA-DR2, supports the hypothesis of genetic predisposition in EBA [27].

The absence of follow-up on our patient represents a limitation of this study, as we are unable to evaluate the therapeutic response. However, we hypothesize that the patient would likely have sought further care had the disease remained or deteriorated.

## 5. Conclusions

While mechanobullous EBA predominantly affects adults, this case study illustrated the condition in a prepubertal male. The treatment with clobetasol propionate in an ointment on the affected skin areas, along with lifestyle modification, was consistent with suggestions that a conservative treatment approach involving topical glucocorticosteroids may be sufficient in mild to moderate pediatric cases. The prognosis for pediatric patients appears to be more favorable compared to that for adults, indicating that systemic therapy should only be considered in more severe or refractory cases. The currently unknown pathogenetic dissimilarities between adult and pediatric cases of the condition may contribute to the different therapeutic outcomes observed in these patient populations, highlighting the need for additional research in this area. Importantly, EBA commonly co-occurs with other autoimmune conditions, necessitating careful monitoring by clinicians. Further studies and case reports are essential to establish evidence-based guidelines for the management of this rare yet impactful disease in pediatric patients.

## Figures and Tables

**Figure 1 antibodies-14-00034-f001:**
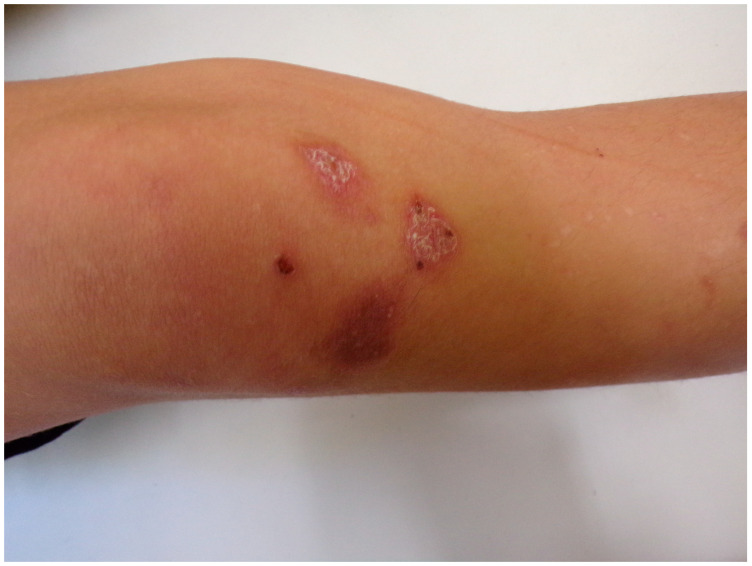
The clinical presentation of the child with EBA was characterized by distinct skin manifestations. Areas of scarring, erosions, and crusting were observed specifically on the right knee. These lesions exhibited a history of blister formation and subsequent rupture, leading to the development of painful erosive wounds. This combination of symptoms highlights the challenging nature of managing mechanobullous EBA in pediatric patients, given the potential for discomfort and secondary infections associated with these skin changes.

**Figure 2 antibodies-14-00034-f002:**
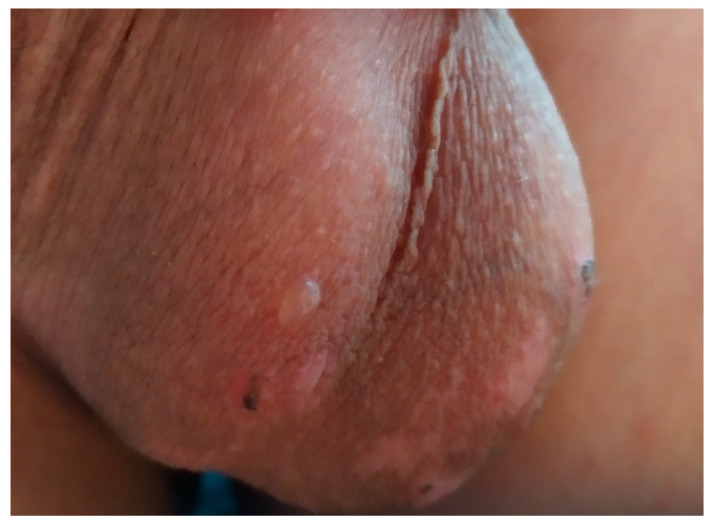
A blister and several prominent scars were visible on the surface of the scrotum, suggesting a history of irritation or injury. The surrounding scars varied in color and texture, hinting at previous skin trauma that had not fully healed.

**Figure 3 antibodies-14-00034-f003:**
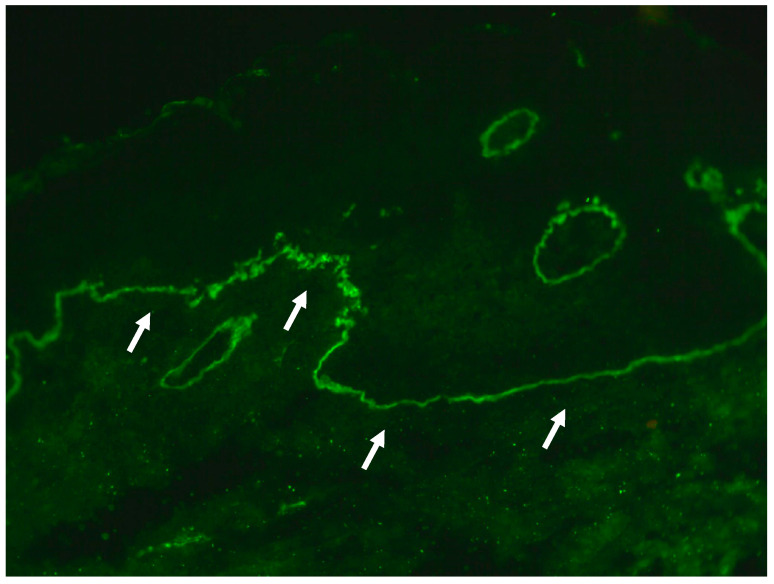
Immunopathological findings. Linear C3 (+++) deposits (arrows) were detected at the dermal–epidermal junction of perilesional scrotal skin in DIF visualized with short-arc mercury lamp-operated microscopy (original objective magnification ×40).

## Data Availability

All data are available upon request.

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
