# Peer review of "An Exceedingly Rare Case of Mechanobullous Epidermolysis Bullosa Acquisita in a Prepubertal Child: A Review of the Clinical and Laboratory Considerations"

_2073-4468, 2025, doi:10.3390/antib14020034_

Round 1
Reviewer 1 Report
Comments and Suggestions for Authors
It is a great pleasure to have the opportunity to review this interesting case report. This manuscript presents a rare case of mechanobullous epidermolysis bullosa acquisita (EBA) in an 11-year-old male football player. It includes a thorough description of the patient’s clinical presentation, diagnostic approach, and treatment. This case report is well-written and fascinating. My comments and questions are as follows.
- In the introduction, an explanation of four clinical subtypes of inflammatory type of EBA is beneficial for physicians to further understand EBA.
- In the case reports, erythema is observed around the scar and crusts. Although the blisters are distributed in the areas prone to trauma, the inflammatory type is not entirely excluded. How did the authors evaluate the inflammation in this case?
- While the authors state that HE staining would only show non-specific findings, histopathological examination would provide additional information helpful in the evaluation of AIBD, such as the type and extent of inflammation, the levels of tissue separation (subepidermal or intraepidermal), the presence or absence of spongiosis and necrotic/apoptotic cells, and so on. Therefore, authors should reconsider the significance of HE staining in AIBD.
- The reason for the negativity of IgG deposition in DIF should be discussed in more detail.
- While ELISA confirmed elevated IgG antibodies against type VII collagen in this case, did the authors investigate the targeted epitope?
- The patient was treated with high-potency topical corticosteroids and lifestyle modification. Given that some pediatric EBA cases require systemic therapy (e.g., prednisone and dapsone), under what circumstances would the authors consider systemic treatment?
Author Response
Dear Reviewer,
Thank you very much for reviewing our manuscript. We sincerely appreciate your time and effort. Please find our response to your comments below written in red.
- In the introduction, an explanation of four clinical subtypes of inflammatory type of EBA is beneficial for physicians to further understand EBA.
We agree with this comment and have included a description of the clinical subtypes in the introduction in lines 41-57. - In the case reports, erythema is observed around the scar and crusts. Although the blisters are distributed in the areas prone to trauma, the inflammatory type is not entirely excluded. How did the authors evaluate the inflammation in this case?
The active lesions were seen just on the scrotum. There, no inflammation was visible. The scars on the knee were indeed surrounded by residual inflammation; we attributed it rather to the damage of the skin by trauma caused by the physical activity of the patient but not by the disease itself.
- While the authors state that HE staining would only show non-specific findings, histopathological examination would provide additional information helpful in the evaluation of AIBD, such as the type and extent of inflammation, the levels of tissue separation (subepidermal or intraepidermal), the presence or absence of spongiosis and necrotic/apoptotic cells, and so on. Therefore, authors should reconsider the significance of HE staining in AIBD.
Indeed, the finding of the subepidermal blister and the neutrophil-dominant dermal inflammatory infiltrate would correspond to the diagnosis. Nevertheless, we wanted to avoid further traumatizing the delicate scrotal skin of a child, so the DIF enabling the detection of autoimmune disease had to be chosen. We have included this information in our manuscript in line 157. - The reason for the negativity of IgG deposition in DIF should be discussed in more detail.
The lack of immunoglobulin detection in DIF might be explained by the possibility that their deposition was below the threshold of the method used. We have included this information in the manuscript in line 112. - While ELISA confirmed elevated IgG antibodies against type VII collagen in this case, did the authors investigate the targeted epitope?
The multiplex ELISA contains the NC1 domain of type VII collagen. This information is included in section 3.2. - The patient was treated with high-potency topical corticosteroids and lifestyle modification. Given that some pediatric EBA cases require systemic therapy (e.g., prednisone and dapsone), under what circumstances would the authors consider systemic treatment?
The systemic glucocorticosteroids were not used because the activity of the disease was mild, covering just the isolated areas of the skin. Importantly, the growth-impairing action of these compounds makes it a much more dangerous therapy option in a prepubertal child. Therefore, systemic treatment should only be considered when the disease does not respond to topical therapy (section 3.2.)
Reviewer 2 Report
Comments and Suggestions for Authors
The manuscript presents a case of mechanobullous epidermolysis bullosa acquisita (EBA) in a prepubertal child, a condition that is exceedingly rare and poorly documented in the literature. With only seven pediatric cases of mechanobullous EBA reported globally to date, this case study contributes novel insights to the field of dermatology and pediatric autoimmune diseases. The discussion on diagnostic approaches and treatment options is relevant to clinicians and researchers, addressing an area of unmet need for evidence-based guidelines in pediatric EBA. The manuscript also highlights the importance of differential diagnosis in autoimmune blistering diseases, which is a critical topic in dermatological practice.
The methods employed in this study are appropriate for the aims of the manuscript. The diagnosis was supported by a robust combination of clinical, histopathological, and immunological investigations. However, the lack of a biopsy for hematoxylin and eosin (H&E) histopathology, while justified as non-essential in this case, may limit the completeness of the diagnostic approach.
The conclusions drawn in the manuscript are generally well-supported by the data presented. The authors convincingly argue for the utility of conservative management (topical glucocorticosteroids and lifestyle modifications) in mild pediatric cases of mechanobullous EBA. They also emphasize the importance of considering EBA in the differential diagnosis of pediatric blistering diseases and the need for further research into pediatric-specific diagnostic and treatment guidelines. However, the manuscript would benefit from a more detailed discussion on the potential limitations of the study, of which the major one is that the patient being lost to follow-up, which prevents evaluation of long-term treatment outcomes.
Minor Points
1. Ensure consistent use of terms such as "mechanobullous EBA" versus "epidermolysis bullosa acquisita" throughout the manuscript.
2. Figures 1 and 2 lack detailed captions that would enhance understanding.
Major Points
1. The lack of follow-up data significantly limits the ability to draw conclusions about the long-term efficacy of conservative treatment. The authors should address this limitation explicitly in the discussion.
2. The manuscript would benefit from a more detailed comparison of treatment outcomes between pediatric and adult cases, particularly regarding the immunological differences (e.g., autoantibody profiles) that are briefly mentioned. This could include a table summarizing key differences reported in the literature.
3. The discussion of clinical implications could be expanded to include recommendations for monitoring pediatric EBA patients for co-occurring autoimmune conditions, such as inflammatory bowel disease, given the reported associations.
4. While the authors advocate for a three-tiered diagnostic approach, they should discuss potential pitfalls or limitations of this methodology, particularly in resource-limited settings where access to multiplex ELISA may be restricted.
1. Minor typographical errors should be corrected (e.g., “lamina propriac” should be “lamina propria”).
Author Response
Dear Reviewer,
Thank you very much for reviewing our manuscript. We sincerely appreciate your time and effort. Please find our response to your comments below written in red.
- Ensure consistent use of terms such as "mechanobullous EBA" versus "epidermolysis bullosa acquisita" throughout the manuscript.
We have revised and corrected the language of our manuscript. - Figures 1 and 2 lack detailed captions that would enhance understanding.
We have made the captions of both figures more detailed and descriptive. - The lack of follow-up data significantly limits the ability to draw conclusions about the long-term efficacy of conservative treatment. The authors should address this limitation explicitly in the discussion.
The fact that the patient could not be followed up is indeed a drawback of our study, as the response to therapy cannot be assessed. Nevertheless, we suspect that the patient would have returned if the disease had persisted or worsened. We have included this information in the discussion section in line 191. - The manuscript would benefit from a more detailed comparison of treatment outcomes between pediatric and adult cases, particularly regarding the immunological differences (e.g., autoantibody profiles) that are briefly mentioned. This could include a table summarizing key differences reported in the literature.
The purpose of the case report was to provide insights into the pediatric EBA. We believe that a more detailed revision of adult EBA would impair the understanding of the information and advice regarding the main topic and would confuse the readers. - The discussion of clinical implications could be expanded to include recommendations for monitoring pediatric EBA patients for co-occurring autoimmune conditions, such as inflammatory bowel disease, given the reported associations.
The information about co-existing autoimmune conditions is included in the manuscript, alongside the possible explanation of their etiopathogenesis. Routine screening in the lack of clinical symptoms is unnecessary. Detailed anamnesis with both children and their parents is enough to evaluate the risk in most cases. - While the authors advocate for a three-tiered diagnostic approach, they should discuss potential pitfalls or limitations of this methodology, particularly in resource-limited settings where access to multiplex ELISA may be restricted.
The diagnostic approach described in our manuscript is essential and in each case of suspected EBA, a patient should be referred to a department disposing of ELISA and DIF.
Reviewer 3 Report
Comments and Suggestions for Authors
The case is very interesting because of its rarity and the authors have provided a well written case report. There is a useful clinical message and advise on the diagnosis of the condition.
I have only very minor comments:
Would you consider mentioning which of the 2 (mechanobullous opr immunological) has the better prognosis?
In Fig. 1 you define the condition as mechanobullous, in the text you describe antibodies.
How often (per day) and how long was clobetasol used until response and until full remission?
How much time has elapsed from diagnosis to now? Has the patient had relapses? How long is the follow up per today? It would be useful if you have this data and add to the paper
Author Response
Dear Reviewer,
Thank you very much for reviewing our manuscript. We sincerely appreciate your time and effort. Please find our response to your comments below written in red.
1. Would you consider mentioning which of the 2 (mechanobullous opr immunological) has the better prognosis?
We have added the information that from our experience, inflammatory phenotype has a better long-term prognosis in line 181.
2. In Fig. 1 you define the condition as mechanobullous, in the text you describe antibodies.
Indeed, it was a mechanobullous variety of EBA, which is mediated by antibodies to type VII collagen, as described in the manuscript.
3. How often (per day) and how long was clobetasol used until response and until full remission?
It was recommended to use it twice daily until achieving healing of lesions (added in line 101), but the response to therapy was impossible to judge as the patient was lost to the follow-up.
4. How much time has elapsed from diagnosis to now? Has the patient had relapses? How long is the follow up per today? It would be useful if you have this data and add to the paper
It has been almost 2 years since the diagnosis. As mentioned in the manuscript, our patient was lost to follow-up. Nevertheless, we suspect that the patient would have returned if the disease had persisted or worsened – we included this information in the discussion section (line 191).